## Registered report

neuroscience/psychology

system-justifying ideologies, right-wing authoritarianism, social dominance orientation, voxel-based morphometry, grey matter volume

**Author for correspondence:**
Gianluca Esposito
e-mail: gianluca.esposito@unitn.it

# Neuroanatomical correlates of system-justifying ideologies: a pre-registered voxel-based morphometry study on right-wing authoritarianism and social dominance orientation

Jan Paolo M. Balagtas[1], Serenella Tolomeo[2],
Bindiya L. Ragunath[1], Paola Rigo[3], Marc H. Bornstein[4]
and Gianluca Esposito[5]

[1]Psychology Program, School of Social Sciences, Nanyang Technological University, Singapore
[2]Institute of High Performance Computing, A*Star, Singapore
[3]Department of Developmental and Social Psychology, University of Padova, Padova, Italy
[4]Child and Family Research, Eunice Kennedy Shriver National Institute of Child Health and Human Development, Bethesda, USA
[5]Department of Psychology and Cognitive Science, University of Trento, Rovereto, Italy

GE, 0000-0002-9442-0254

System-justifying ideologies are a cluster of ideals that perpetuate a hierarchical social system despite being fraught with inequalities. Right-wing authoritarianism (RWA) and social dominance orientation (SDO) are two ideologies that have received much attention in the literature separately and together. Given that these ideologies are considered to be stable individual differences that are likely to have an evolutionary basis, there has yet to be any examination for volumetric brain structures associated with these variables. Here, we proposed an investigation of overlapping and non-overlapping brain regions associated with RWA and SDO in a sample recruited in Singapore. Indeed, it will be interesting to determine how RWA and SDO correlate in a country that proactively promotes institutionalized multi-culturalism such as Singapore. RWA and SDO scores were collected via self-report measures from healthy individuals (39 males and 43 females; age 25.89 ± 5.68 years). Consequently, voxel-based morphometry (VBM) whole brain and region of interest

(ROI) analyses were employed to identify neuroanatomical correlates of these system-justifying ideologies. RWA and SDO scores were strongly correlated despite the low ideological contrast in Singapore's sociopolitical context. The whole brain analysis did not reveal any significant clusters associated with either RWA or SDO. The ROI analyses revealed clusters in the bilateral amygdala and ventromedial prefrontal cortex (vmPFC) that were associated with both RWA and SDO scores, whereas two clusters in the left anterior insula were negatively associated with only SDO scores. The study corroborates the claim of RWA and SDO as stable individual differences with identifiable neuroanatomical correlates, but our exploratory analysis suggests evidence that precludes any definitive conclusion based on the present evidence.

# 1. Introduction

## 1.1. Background

Altemeyer [1] conceptualized right-wing authoritarianism (RWA) as an ideology that can be understood as a cluster of three covarying traits: authoritarian submission, authoritarian aggression and conventionalism. That is, these traits comprise a singular measure of RWA. Authoritarian submission or the tendency to almost unquestioningly obey an authority figure is one such hallmark trait of RWA. The field of social psychology has also empirically investigated this notion of obedience to authority through controlled manipulations in the laboratory. Arguably, no one is more influential in this regard than Stanley Milgram and his obedience experiments. In these seminal studies, Milgram [2,3] found that more than half of his 'teacher' participants were willing to deliver (allegedly) fatal electric shocks of 450 V to a 'learner' (a research confederate) on the other side of the room with nothing more than a reassuring you-will-not-be-held-responsible statement by a man in a white coat. These were certainly not demented nor sadistic individuals who participated. They were average university students and adults from the community [3]. It is often interpreted from these experiments that the influence of an authority figure is strong enough for individuals to forgo their principles to follow instructions regardless of the harm they may inflict on another. In fact, these findings have been extended to other countries beyond the Americas, such as Jordan [4] and Australia [5], indicating that this finding is not culture specific (see [6] for review). Nonetheless, one can easily dismiss this as a peculiarity of that period of time. Maybe people were just more 'authoritarian' then. On the contrary, nearly half a century after the first series of Milgram's experiments, Burger [7] replicated—albeit not completely, given the ethical issues rife in the original study—the same obedience task and drew the conclusion that typical individuals respond to authority much the same way now as they did 45 years ago. The replication of these findings across generations suggests that authoritarian submission, and by extension the RWA ideology or 'follower's authoritarianism', may have a strong biological component (see [8]). A similar case can be made with a related ideology known as social dominance orientation (SDO), which has conversely been referred to as 'leader's authoritarianism' [9]. We propose that the biological component of RWA and SDO may manifest as neural structures that differ across individuals. Therefore, in this report, we aim to use voxel-based morphometry (VBM) to identify the neuroanatomical correlates that vary as a function of self-report scores in RWA and SDO scales.

## 1.2. System-justifying ideologies

Around the globe, the majority of modern societies are grounded on a hierarchical social system with a substantial proportion of its members found to be tolerant, or sometimes accepting, of the structural inequalities that come about as a result of the hierarchy [10–13]. Notably, there is an uneven distribution of resources among its members, whereby some have more than others—often substantially more—yet people still believe that these are distributed fairly. Inherent in such hierarchies are inequalities in the treatment of its members by virtue of race, gender, political orientation and other traits that may characterize such individuals as the 'outgroup' [14–16]. Despite this recognizable unequal treatment, society members, particularly those at the bottom of the hierarchy [17,18], have a general tendency to legitimize and accept the status quo [10,19,20].

Jost & Hunyady [21] argue that individuals adopt different system-justifying ideologies as a means to maintain the status quo. These include, but are not limited to, meritocratic ideology, political conservatism, belief in a just world, RWA and SDO. These ideologies serve the same purpose of rationalizing and

legitimizing social inequalities, but the observable manifestations of each ideology may differ. Among many others, the last two are prominent ideologies that are worth highlighting: RWA and SDO. We examine these two specifically for three reasons. First, there is a substantial amount of existing research on these two ideologies, both separately and simultaneously [22–24]. Second, RWA and SDO, when pitted against other relevant predictors, have been shown to account for the vast majority of variance in intergroup attitudes and behaviours important in perpetuating a hierarchical social system, such as prejudice (RWA: [1,25,26]; SDO: [27,28]; RWA & SDO: [29–33]). Third, past cross-cultural research documents a correlated but independent relationship between RWA and SDO [22,24], which presents a reasonably robust foundation for examination beyond analysis of explicit evaluations (i.e. self-report). Before going further, we first need to expound on these two measures.

## 1.3. Right-wing authoritarianism

RWA is indexed as a self-report score based on a 32-item questionnaire constructed by Altemeyer [1,34,35] to measure the covariation in an individual's (i) submission to authority (i.e. authoritarian submission), (ii) aggression against those who deviate from the norm (i.e. authoritarian aggression), and (iii) preservation of traditions, typically those advocated by authority figures (i.e. conventionalism). Each item is rated on a 9-point Likert scale indicating the individual's agreement with each statement from −4 (very strongly disagree) to +4 (very strongly agree). Although recent research argues that dividing RWA into its subcomponents can add greater nuance and better explanatory power to intergroup outcomes (e.g. authoritarianism–submission–traditionalism scale [36]; 'authoritarian aggression and submission' and 'conservatism' dimensions [37]), past findings involving correlations with the superordinate RWA measure support the notion that this singular measure is likely to be more than sufficient in capturing the large majority of variance in the data [1,22,38].

## 1.4. Social dominance orientation

SDO is also measured using a self-report score based on a 16-item questionnaire [27]. Each item is rated on a 7-point Likert scale indicating the individual's agreement with each statement from 1 (strongly disagree) to 7 (strongly agree). Pratto *et al.* [27] define SDO as 'the extent to which one desires that one's in-group dominate and be superior to out-group'. One of the most notable ways by which individuals justify inequalities in society is by legitimizing the discrimination in the existing hierarchy as indisputable facts [39]. Following this reasoning, the social dominance model predicts those higher in SDO scores are more likely to endorse these hierarchy-legitimizing beliefs and the policies that maintain these beliefs. Indeed, this conjecture has been examined and verified in various extensions of the original study ([27]; e.g. [40,41]).

## 1.5. Right-wing authoritarianism and social dominance orientation as separate constructs

RWA and SDO represent relatively stable and longstanding individual differences. In other words, people vary in their levels of RWA and SDO. These differences tend to decrease with age but show reasonable consistency in adult samples [42]. Despite being measured as separate scales in most research, some have raised the possibility that the correlation between RWA and SDO might denote that they can be subsumed to a uni-dimensional spectrum [43,44]; for opposing views, see [30,45]. Conceptually, this makes sense given that both RWA and SDO are system-justifying ideologies that strive to directly influence the stability of the social hierarchy (e.g. through promoting prejudice attitudes and behaviour towards low-ranking members to keep them at the bottom), as opposed to others, such as Protestant work ethic and meritocracy, that do so indirectly (e.g. espouses the idea of those who are at the bottom to have deserved their situation; [21]). Moreover, their authoritarian nature—both measures deal with either submitting or subjecting others to authority and thus maintaining a hierarchical social structure—can lead one to suspect that RWA and SDO exist along a single ideological dimension.

Nonetheless, there is much more evidence to indicate that these measures reflect independent constructs. Roccato & Ricolfi [24] conducted a cross-cultural meta-analysis to show that the correlation between the two measures is contingent on the ideological contrast of the country by which the population was studied. For instance, Italy has a distinctively left–right political environment, in which case RWA and SDO are likely to covary due to the more structured and rigid political attitudes and behaviours anchored along the left–right dimension. Individuals who are leftist (i.e. liberal) were

likely to score low on both RWA and SDO, whereas rightist individuals (i.e. conservative) scored high on both RWA and SDO, particularly among those with a strong identification with their political orientation [23]. Conversely, countries with minor ideological contrast, that is, political attitudes and behaviours do not align very neatly along a left–right dimension, reported RWA and SDO independent of each other. Indeed, one study in a Canadian sample (a country low in ideological contrast) reports one of the lowest correlations between RWA and SDO (e.g. $r = 0.08$; [1]). A similar case can be said about the phenomenon that both RWA and SDO reliably and, in most cases, in the same direction, predict prejudice. The dual process model (DPM) by Duckitt [46] asserts that, although the attitudinal and behavioural outcomes are visibly similar, the underlying beliefs for prejudice in high SDO and RWA individuals are different. According to the model, prejudice attitudes and behaviour in high RWA individuals are motivated by threats to one's security and unpredictability whereas high SDO individuals are motivated by superiority and competitiveness [22,29,30,33,38,47–49]. The DPM model also accounts for the systematic cross-cultural variations by the degree of ideological contrast in each country [24].

However, a study by Roets et al. [25] in Singapore represents a unique instance where higher RWA does not predict greater prejudice. Individuals who reported higher RWA scores also reported more positive attitudes toward outgroup individuals, contrary to the predicted direction by the DPM. It is worth noting that this relationship was mediated by the individuals' perception of the government's stance on multi-culturalism in Singapore. To elaborate, the Singapore government enacts measures to promote intergroup contact and interaction among its citizens, such as through ethnic quotas in public residential buildings and schools [50]. The top-down influence of the government has, in this case, reversed the typically positive correlation between RWA and prejudice. To reiterate, the correlation of RWA and SDO is largely a function of the ideological contrast of the national context it is embedded in [24]. Specifically, countries that have a strong ideological contrast tend to promote a political left– right dimension such that individuals that ascribe to a specific political ideology (i.e. left versus right) tend to show positively correlated RWA and SDO scores. More relevant to this paper, the sociopolitical context of Singapore does not feature an explicit left–right dimensionality and so, would be considered a country of low ideological contrast. Therefore, the DPM model would predict that RWA and SDO would be largely independent of each other in the Singapore context. However, the DPM does not make clear predictions with regard to institutionalized processes (i.e. governmental promotion of multi-culturalism) that may influence system-justifying ideologies (see [51]). To the best of our knowledge, no study has yet been done to investigate the correlation between RWA and SDO in Singapore. Though the influence of multi-culturalism policy on SDO has not yet been measured in Singapore, research in other countries that show a negative correlation between support for multi-culturalism and SDO indicates a trend towards that direction (e.g. France [52]; USA [53]). In fact, the study conducted by Levin et al. [53] demonstrated multi-culturalism as a strong mediating factor for the relationship between SDO and prejudice towards minority groups. Therefore, based on the large role multi-culturalism plays in Singapore's sociopolitical context, there is good reason to believe that RWA and SDO will be influenced similarly, thus boosting the correlation between these two ideologies (H1). Notably, this would be in stark contrast to other industrialized and low ideologized nations identified in the meta-analysis by Roccato and Ricolfi [24] that feature weak correlational measures of RWA and SDO.

To proceed with investigating the neuroanatomical correlates of RWA and SDO with a strong predictive framework, we believe the core theoretical stance we prescribe bears repetition. Though it is unlikely that RWA and SDO measure a uni-dimensional construct, it is instructive for this investigation to respect them as system-justifying ideologies with the same goal of maintaining existing social hierarchies albeit achieved in different ways. Therefore, we postulate that the status of RWA and SDO as system-justifying ideologies will manifest as an overlap in at least one brain region. At the same time, the DPM model argues that RWA and SDO are derived from different underlying motivations and observable outcomes (i.e. the different ways existing social hierarchies are maintained). In alignment with this model, we predict that RWA and SDO will also correlate with at least one other brain region independent of one another.

## 1.6. Neural correlates of right-wing authoritarianism and social dominance orientation

This paper aims to determine the neuroanatomical correlates of RWA and SDO. In particular, we aim to distinguish the overlapping and non-overlapping brain regions associated with scores in the RWA and SDO scales using a voxel-based morphometry (VBM) technique. Although traditional self-report measurements of RWA and SDO have demonstrated robust reliability and validity across multiple

studies, the examination into the neural bases of RWA and SDO can provide more solid evidence for their status as stable individual differences. Recently, there have been some efforts toward this direction. For instance, the DPM of threat and competition has been shown to map well with the two-dimensional evolutionary framework of economic conservatism (i.e. dominance) and social conservatism (i.e. authoritarianism) such that RWA and SDO are likely to have been adaptive and have, over evolutionary time, developed a heritable biological basis [38]. This evolutionary framework can help navigate the possible neural substrates of RWA and SDO, and in particular, where these two system-justifying ideologies intersect in the brain (i.e. overlapping brain regions). We also predict that these overlapping and non-overlapping brain regions correspond to the tenets of DPM. We predict that RWA and SDO would involve identical brain regions as they are both system-justifying ideologies that individuals espouse to maintain the hierarchical structure of society. Additionally, these constructs correlate but are nonetheless independent, and would therefore recruit unique brain regions to differentially substantiate these ideologies in terms of antecedents and outcomes as propounded by the DPM model.

Granted the conceptual overlap and correlation in RWA and SDO, we have reasonable confidence that these two ideologies are likely to covary with, at least, one identical brain region. The amygdala is a potential candidate as an overlapping neuroanatomical correlate (see [54]). This subcortical region has been demonstrated to be involved in a wide range of psychological processes, such as outgroup discrimination [55], political inclinations [56,57] and even social network size [58]. Though seemingly disparate constructs, these processes interact as socially relevant facets of any large society [59], and the evidence surmounting the recurring relevance of the amygdala suggests an overarching social process governed by this region. In fact, research in how amygdala volume correlates with scores in the system justification scale points to its important role in the maintenance of hierarchical social systems [60]. Therefore, both RWA and SDO, as system-justifying ideologies, are also associated with perpetuating societal hierarchies [21], and hence, the scores in these measures will presumably also correlate positively with amygdala volume (H2).

In terms of non-overlapping regions, the neurological research in RWA and SDO use different paradigms. One line of work in the neural substrate of RWA lies in neuropsychological research [8,61]. Asp *et al.* [61] have shown that the ablation of the ventromedial prefrontal cortex (vmPFC) led to greater endorsement of RWA. This association of RWA with the vmPFC lines neatly with the false tagging theory (FTT) that posits that the vmPFC is involved in assessing the veracity of a belief [62]. In other words, the vmPFC underpins the ability to doubt a belief. Predictably, individuals with damage to their vmPFC are more likely to subscribe to fundamentalist beliefs because of this 'doubt deficit'. Similarly, individuals with high RWA are also likely to rely on authority to dictate their beliefs and are often rigid and immutable to evidence [1,63]. In line with these associations, damage to the vmPFC would lead to increased RWA scores, which was supported by their findings [61]. The authors of the study also went on to further explicate the distinction between religious fundamentalism and RWA, which can be easily conflated due to the nature of large roles of religious authority figures in promoting RWA [26,33]. Importantly, after partialling out religious fundamentalism, Asp *et al.* [61] found that the degree of vmPFC damage still led to higher RWA scores, suggesting that the fundamentalist beliefs associated with RWA is not exclusive to a religious context [64]. In other studies, vmPFC lesions are also implicated in changes in performances of tasks related to the covariations of RWA, such as authoritarian submission (e.g. tests of belief; [65]) and conventionalism (e.g. gender stereotyping; [66]), further corroborating the role of vmPFC in RWA. These findings implicating vmPFC damage to increase in self-report RWA and changes in its covariations suggests that this brain region may have a volumetric relation with individual difference in RWA. That is, we hypothesize that variation in RWA scores will be negatively associated with the structural volume of vmPFC (H3).

In terms of SDO, although no current work has been done to directly examine its neuroanatomical correlates, research in functional brain imaging suggests some possible areas of interest [67–69]. It is important to highlight that these functional scans reveal varying activated brain areas. These variations in brain areas are potentially attributable to the different tasks employed in each study. Chiao *et al.* [68] measured empathic response to stimuli involving pain (and no pain) and found higher SDO scores covaried with reduced activation in the anterior cingulate cortex (ACC) and insula when perceiving pain in others. To rule out alternative explanations, the authors also conducted a thorough manipulation check on whether the participants think that the individual in the image is experiencing pain. By contrast, a more recent study found that SDO was associated with activity in the dorsolateral prefrontal cortex (dlPFC) and superior temporal sulcus (STS) when viewing faces associated with different social ranks [69]. The latter study, however, defined 'social ranks' based on rate of winning in the task, operating under the presumption that those who win more than the participant are seen as 'superior', and those who lose more as 'inferior'. It is possible that participants did not perceive any social ranking

during the task at all. As there was no manipulation check for this implicit assumption, it is not clear why SDO scores covaried with dlPFC and STS activity. Consequently, the association between SDO and dlPFC and STS regions may not be borne out once this particular task is no longer carried out during the brain scan. Additionally, another study by Baumgartner et al. [70] did not find a significant association between subscales of SDO and volume of dorsolateral medial PFC, a region anatomically proximal to the dlPFC (see electronic supplementary material, S3). It is thus unlikely that SDO will be associated with this cluster of regions. The methodology in the study by Chiao et al. [68] more conceptually aligns with the construct of social dominance measured by the SDO scale because it involves empathic concern, a process that has an important relationship with social dominance [71]. Another study involving the role of SDO in moderating the performance of individuals in a gaze-following task using left- and right-wing politicians' images as stimuli implicated the midcingulate cortex and left anterior insula, among other related regions— increased SDO was negatively associated with activation in these regions [67]. Both studies suggest that there is at least one brain area that is reliably associated with SDO and that it is not simply a consequence of utilizing a peculiar task. Therefore, we believe it is likely that variation in SDO scores will be negatively associated with structural volume of the left anterior insula (H4).

Our prediction of non-overlapping neuroanatomical regions associated with RWA and SDO suggests that these two ideologies also recruit separate brain regions that reflect the different underlying beliefs that underlie RWA and SDO as predicted by the DPM model. Though there is no direct evidence for this double dissociation, some indirect evidence in the literature hints to this possibility. The study by Asp et al. [61] demonstrated that only damage to vmFPC was significantly associated with higher RWA scores compared with healthy controls. Patients with damage to other neural structures, including those that are involved with emotion, did not show this increase in RWA scores. The etiologies of these non-vmPFC lesions were not overly specified. Notwithstanding, this distinctiveness of RWA scores associated with only vmPFC damage and not other cortical regions implicated in emotional processing leads us to hypothesize that RWA is likely not to associate with the left anterior insula. Importantly, this non-vmPFC lesion group excludes patients with specific damage to the amygdala. Thus, the predicted overlapping association of RWA and SDO with the amygdala remains intact. Moreover, only Chiao et al. [68] have thus far conducted a whole-brain analysis to identify regions that covary with SDO scores during a functional magnetic resonance imaging (fMRI) task. SDO scores were a significant predictor of frontal areas, namely, inferior, superior and middle frontal gyri activity, in addition to the aforementioned ACC and insula activity when participants engaged in an empathic task. However, after controlling for age and self-reported dispositional empathy, only the ACC and left anterior insula were left as regions significantly associated with SDO scores. To our knowledge, no other studies have conducted a whole-brain analysis involving SDO. Comparing with the study by Cazzato et al. [72], only the left anterior insula region consistently covaries with SDO scores across different fMRI tasks. Based on the limited research on this topic, we hypothesize that SDO but not RWA will be associated with the left anterior insula, and RWA but not SDO will be associated with vmPFC.

## 1.7. Present study

Both RWA and SDO have been shown to exhibit cross-cultural validity [24], temporal stability 73], and association with brain lesion (RWA [61]) and brain activity (SDO [67,68]). The neurological research on authoritarian-related and dominance-related ideologies has laid the groundwork on potential neuroanatomical correlates of SDO and RWA and their possible overlaps. Moreover, the evolutionary framework provides evidence for the adaptive potential of a biological basis for RWA and SDO. Therefore, it is reasonable to suspect that the individual difference in scores for these measures would covary with neuroanatomical differences (see [74]). Yet, these neuroanatomical correlates have not been examined. In addition, we will also assess the relationship between RWA and SDO in the Singapore population. It is worth noting that the sample represented in this study is only of the majority Chinese ethnic group. In a review of the influence of policies on intergroup relations, Guimond et al. [51] assert that the impacts of intergroup-related policies, such as multi-culturalism policies, vary to the extent such that cultural perceptions of majority group individuals towards such policies may not represent those of minority group individuals. Therefore, findings in this paper may not extrapolate to the entire Singapore population and may be restricted to those of the ethnic majority group. Nonetheless, the study still holds merit in that limiting the study to only the majority ethnic group can allow the focus on a relatively homogeneous sample to maximize statistical power. Future studies that examine neuroanatomical correlates of RWA and SDO scores should subsequently include ethnic minority groups to examine possible differences in their neuroanatomical correlates of RWA and SDO.

Notwithstanding, we put forward the following specific hypotheses we aim to address: (i) H1: RWA and SDO will be strongly correlated despite the low ideological contrast in Singapore's sociopolitical context, (ii) H2: volume of the amygdala will be positively associated with both RWA and SDO scores, (iii) H3: vmPFC volume will be negatively associated with only RWA, and (iv) H4: left anterior insula volume will be negatively associated with only SDO. To verify these predictions, we will conduct a correlational test for (a) and VBM analyses to estimate grey matter volume in these brain regions for (ii)–(iv).

In this proposed study, we will be conducting secondary analysis of structural MRI data of participants that were recruited for an fMRI study. To do so, we will be focusing on the structural MRI images that were used as an anatomical reference for that study. Responses to RWA and SDO scales were also collected from these participants but these have not been analysed thus far. Taken together, the proposed statistical analyses will primarily consist of a correlational test between RWA and SDO scales, and VBM analyses [75] of prespecified brain regions implicated in RWA, SDO or both. We opted for a region of interest (ROI) approach to strengthen the power to detect true effects in our study [76]. The ROIs identified in our analyses are primarily derived from the regions specified in hypotheses (ii)–(iv), which were in turn informed by our review of the existing literature. In addition to the ROI analyses, a whole-brain analysis will also be conducted.

# 2. Methods and materials

## 2.1. Overview

The self-report and structural MRI data collection has already concluded. However, the raw self-report data have not been analysed prior to this proposal. In this study, we aim to analyse the RWA and SDO data and test our hypotheses by conducting a combined ROI and whole-brain VBM analysis. Additionally, we will test the correlation in self-report scores of RWA and SDO scales.

## 2.2. Participants

Ninety-one (46 females) participants were recruited for this study (age range is 21 to 41 years old). All participants were Singaporean Chinese students recruited from Nanyang Technological University (NTU) or Singaporean Chinese adults from the local community. Each participant was required to provide written informed consent before participating in the study in accordance to the Declaration of Helsinki. This study was approved by the NTU IRB (protocol 2017-01-029).

There were two phases of recruitment in the study. The inclusion criteria of both phases are as follows: (i) Chinese ethnicity (ii) English-speaking, (iii) right-handed, (iv) possess normal or corrected-to-normal vision and hearing, (v) no diagnosis of intellectual disabilities, (vi) no psychiatric/neurological illness, and (vii) no history of illicit drug use. Eligible participants were instructed not to consume any caffeine or medication 24 h prior to their scan. We excluded Chinese Singaporean participants who travelled overseas for more than two months over the past six months from the time of the scan session. In addition, with particular focus on the safety of the participants, individuals with claustrophobia, any metallic prosthesis and/or copper intrauterine devices were not eligible for the MRI section of the study. Finally, female participants at any stage of pregnancy were not eligible for the study. The first recruitment collected data from $N = 56$ (27 females; mean age $23.05 \pm 1.31$ years).

The second recruitment phase had identical inclusion and exclusion criteria as the first phase. The main difference is that for this phase, the target sample was middle-aged adults from the community. The second recruitment collected data from $N = 35$ (19 females; mean age $31.59 \pm 6.59$ years).

In total, 91 participants were recruited to participate in this study. Nine participants were excluded from the main analysis because of incomplete MRI or self-report data. Therefore, the final sample is $N = 82$ (43 females; mean age $25.89 \pm 5.68$ years).

## 2.3. Materials

### 2.3.1. Right-wing authoritarianism scale

The right-wing authoritarianism scale (RWA; [34]) is a 32-item self-report scale that measures authoritarianism. The RWA scale has been shown to demonstrate high degrees of reliability and validity [34,35,77]. The RWA scale in this study used a 22-item version ([78]; appendix A). The 22-

item scale is used in this study as it is the most updated version of the RWA scale and demonstrates comparable psychometric properties as the original 32-item scale. Therefore, we opted for this shortened version of the scale. Participants responded to them on a 9-point Likert scale with anchors at each point. The responses ranged from −4 (very strongly disagree) through 0 (neutral) to +4 (very strongly agree). The scale used in this study sampled statements that represent the three covariations in the original 32-item RWA scale, such as 'The established authorities generally turn out to be right about things, while the radicals and protestors are usually just "loud mouths" showing off their ignorance' reflecting authoritarian submission trait, 'Our country will be destroyed someday if we do not smash the perversions eating away at our moral fibre and traditional beliefs' reflecting the authoritarian aggression trait, and 'The "old-fashioned ways" and "old-fashioned values" still show the best way to live' reflecting the conventionalism trait. Though they reflect arguably independent subscales [36,37], the consensus in the literature indicates that total RWA score is sensitive enough to detect variation in authoritarian-related individual differences (e.g. [1,22]). Moreover, dividing either SDO or RWA into their respective subcomponents adds greater complexity and better explanatory power to expected psychological and neurological outcomes, but the correlations between SDO and RWA in past research supports the notion that they are more than likely to be sufficient in capturing the already greater majority of variance in the data [38]. Therefore, a single RWA score was obtained by summing the responses to each item, after reverse-coding the anti-authoritarian items.

At this juncture, it is worth noting that the version of the scale used in this study has one main drawback. As the 22-item version of the RWA scale uses double- or triple-barrelled questions, it is not feasible to tease apart which of the covariations (i.e. authoritarian submission, authoritarian aggression, conventionalism) are reflected in the participants' response to each item. That is, the 22-item RWA scale can only be viewed as a uni-dimensional trait with three underlying covariations. This is important to acknowledge, given that recent research into the structure of RWA demonstrates that the three dimensions can be viewed as separate subscales (e.g. [79,80]). In addition, Arikin & Sekercioglu [81] also argue that the construct of authoritarianism may be better conceptualized as a predisposition as opposed to a stable trait. RWA as a multi-dimensional construct is a direction for future research that is worth exploring, particularly whether or not the three separate subscales map better at a neural level than the superordinate RWA scale. Nevertheless, we have good reason to believe that there is merit in exploring whether RWA—conceptualized as a uni-dimensional trait—is reflected in brain structure. With regard to our hypotheses, the theoretical frameworks we are basing our predictions on (i.e. system-justifying ideologies and DPM model) conceptualize RWA as a singular measure that can be contrasted with SDO, and so to follow how RWA has been studied thus far would allow us to directly test whether or not these frameworks are valid at a neural level. The DPM literature has also employed cross-lagged data to show RWA has acceptable temporal reliability across at least a five-month period, after accounting for the dangerous worldview, which moderates RWA as predicted by the DPM model [29,49]. Moreover, one study conducted a set of factor analyses of RWA (and SDO) showing that both multi-dimensional and uni-dimensional models of RWA demonstrate acceptable fit to response data granted the items themselves were already divided into their respective subscales [82]. Kandler *et al*. used a 12-item variation of the RWA scale with three clearly delineated subscales (and the original 16-item SDO scale with two subscales). Recruiting a large sample ($N = 1437$), the responses of these individuals to the RWA and SDO scales were subjected to confirmatory factor analyses fitting three alternative models. Their findings indicated acceptable fit to the data where RWA was modelled as three factors, one factor and one superordinate factor with three subordinate factors. Overall, the authors concluded that the multi-dimensional models do not provide substantially better fit that warrants opting out of a more parsimonious uni-dimensional conceptualization of RWA (and SDO). We believe this further supports our stance in that even after the removal of items that overlap across the three covarying traits, RWA may be best conceptualized as a unitary construct. This gives credence to the conceptualization we adopted of RWA as a relatively stable uni-dimensional trait in this study.

### 2.3.2. Social dominance orientation scale

The social dominance orientation (SDO; [27]) scale is a 16-item self-report scale that measures social dominance. The SDO scale has been shown to demonstrate high degrees of reliability and validity [27]. The SDO scale in this study used the original 16-item version (appendix B). Participants responded to them on a 7-point Likert scale with anchors at each extreme. The response ranged from 1 (strongly disagree) to 7 (strongly agree). Examples of these statements include, 'Some groups of people are simply inferior to other groups', 'In getting what you want, it is sometimes necessary to use force

against other groups' and 'It's OK if some groups have more of a chance in life than others'. A single SDO score was obtained by summing the responses to each item, after reverse-coding the anti-dominance items.

## 2.4. Self-report data analysis

Prior to running inferential statistical analysis, RWA and SDO scores will undergo preliminary analysis to check that assumptions for parametric testing are fulfilled. In particular, the normality of responses for each item will be verified using skewness and kurtosis recommended cut-off of ±3 values. Items that do not fulfil the normality assumption will be removed from further analysis. To assess the inter-item reliability of the RWA and SDO scale, a Cronbach's alpha will be calculated for each scale. The correlation between RWA and SDO will be measured using a test of Pearson's correlation. Both statistical tests will be performed using SPSS 26.0 (SPSS Inc., Chicago, IL).

## 2.5. Power analysis

In this study, we will conduct a set of ROI analyses based on prespecified brain regions shown to have associations with RWA, SDO or both. Because there has yet to be any published work measuring the neuroanatomical correlates of RWA and SDO in the extant literature, we are unable to make specific estimates for the effect size. Hence, we decided upon a medium effect size of $f = 0.15$ for our analysis. To achieve at least a power value of 0.80 at an alpha level of $p < 0.05$, a sample size of $N = 55$ is needed. In addition, the test of correlation between RWA and SDO requires a sample size of $N = 67$ to achieve a power of 0.8 at $p < 0.05$. Our study exceeds both criteria with a final sample size of $N = 82$, and consequently, we believe this to be a well-powered sample to provide an accurate estimate of the effect size regarding system-justifying ideologies and their respective structural brain associations.

## 2.6. Magnetic resonance imaging acquisition

All MRI data were collected using a Siemens Magnetom Prisma 3-Tesla MRI Scanner with 64-channel head coil. High-resolution T1-weighted MPRAGE sequences (192 slices; TR 2300 ms; TI 900 ms; flip angle 8 degrees; voxel size 1 mm) were obtained to serve as neuroanatomical raw data. Each participant was fixed to an external head restraint to minimize head movement during the scan.

## 2.7. Voxel-based morphometry preprocessing

Processing of the structural data will be performed using Statistical Parametric Mapping (SPM12; Wellcome Department of Imaging Neuroscience, http://www.fil.ion.ucl.ac.uk/spm/software/spm12) on Matlab 2021a platform. First, T1-weighted images will be segmented using the diffeomorphic anatomical registration through exponentiated lie algebra (DARTEL) for intersubject registration into grey matter probability maps. Images are to be spatially normalized with modulation to preserve the total amount of grey matter, then transformed into the Montreal Neurological Institute (MNI) stereotactic space to produce $1 \times 1 \times 1$ mm$^3$ voxels. Finally, they will be smoothed by convolving the images with an isotropic Gaussian kernel of 12 mm full width at half maximum (FWHM; [57,60,83]).

## 2.8. Whole brain analysis

We will be conducting an exploratory whole-brain analysis using the DARTEL package in SPM12. RWA or SDO scores will be used as contrasts to test significance of regressions coefficients from zero value. Significance thresholds will be set at a peak-level threshold of $p < 0.05$ with family-wise error (FWE) correction, and uncorrected voxel-wise level of $p < 0.001$.

In these analyses, we intend to control for total intracranial volume (TIV), age and gender by including them into the regression model as independent 'nuisance' variables. TIV is an important variable to account for particularly in ROI-based volumetric measures because such subtle differences in regional brain volume may be confounded by individual differences in overall brain size [84]. We are also controlling for age not only because TIV varies as a function of age [85], but also because both RWA and SDO have been shown to decrease with age [1,42]. Accounting for age is also necessitated in this study because the analysis will include participants from two different age groups, a young adult sample and a middle-aged adult sample. We would expect both self-report and volumetric brain differences between these two age groups so including age in the regression model

will minimize confounds due to age differences. Finally, past research also suggests a gender difference in self-reports of RWA and SDO. In particular, women tend to report higher RWA scores than men [86], whereas men tend to report higher SDO scores than women [27]. Combined with an overall brain volume difference between men and women [87–89], we reckoned to control for gender would facilitate in identifying significant neuroanatomical correlates, as we predict with the age variable. We would like to emphasize that although system-justifying ideologies and regional (and overall) brain volume do seem to vary with age and gender, these are treated as nuisance variables in the main analysis as they do not comprise the main objectives of the study.

We will investigate the association between grey matter volume (GMV) and scores in the RWA and SDO scales using multiple regression analyses. Each multiple regression analysis will use ordinary least-squares models with smoothed, corrected GMV estimate as the dependent variable, and RWA (or SDO) score, gender, age and TIV as independent variables.

## 2.9. Region of interest analysis

To supplement the exploratory whole-brain analysis, an *a priori* ROI analysis will also be conducted. The ROI analyses will focus on these predictions: the grey matter volume of the amygdala will be positively associated with both RWA and SDO scores (H2), grey matter volume of the vmPFC will be negatively associated with only RWA (H3), and grey matter volume of the left anterior insula will be negatively associated with only SDO (H4). Specifically, our main independent variables of interest are the two measures of system-justifying ideologies (i.e. RWA and SDO scores). These regions were selected based on previous research implicating these respective regions to RWA, SDO or both. With respect to the amygdala, this is the only hypothesis that is based on a previously conducted neuroanatomical study [60]. However, we would like to reiterate that neither the RWA nor SDO scales were analysed in the full sample of the study. Instead, a general system justification scale was used [90]. These items tap on similar beliefs consistent with high RWA (e.g. 'In general, you find society to be fair') or SDO (e.g. 'Society is set up so that people usually get what they deserve'). It is worth noting that Nam *et al.* [60] previously did not find a significant correlation between SDO and amygdala volume. However, the absence of a relationship may probably have been due to the relatively small sample size (N = 37) used to analyse this relationship. Equipped with a more well-powered sample, our study (N = 82) presents a more definitive measure of a neuroanatomical correlate not only of SDO but also RWA. Therefore, we reasoned that RWA and SDO are likely to also correlate with amygdala volume (H2), as does the general system justification scale in the study by Nam *et al*. The link between vmPFC and RWA is also unique among the three hypotheses in that it is the only one based on a set of neuropsychological studies involving neurology patients with a lesion in this brain region [61,62,65]. This points to a specific role of the vmPFC in modulating RWA ideologies, such that damage to this region leads to a manifested change in both RWA scores and outcomes related to RWA such as increased religious fundamentalism [61] and more generally, a magnified susceptibility to misleading information [65]. We believe it is not that far-fetched to suspect that the regional volume in vmPFC correlates with the degree of ascription to RWA (H3). Finally, the left anterior insula was identified primarily from an fMRI study by Chiao *et al.* [68] that found this region to correlate significantly with SDO scores during a pain perception task. This same region was detected in another study using a different task so we can be confident that these correlations are not simply an idiosyncrasy of a specific task type [67]. However, it is worth mentioning that the insula was implicated in this latter study as part of the 'social orienting circuit' in the brain but was not reported to be directly correlated to SDO scores. Nevertheless, both studies taken at face value did use tasks that tap on the essence of SDO—that is the preference for dominance in terms of observing the pain of others [68] or perceived similarity in others [67]. Therefore, based on the limited literature on this topic, we argue that there is value in the present structural ROI-based analysis and we do expect SDO scores to correlate with the regional volume of left anterior insula (H4).

For each participant, we will average voxel-wise GMV values for each ROI individually, which then serves as the dependent variable for our main analyses.

We will investigate the association between GMV and scores in the RWA and SDO scales using ROI multiple regression analyses. As the brain areas in our hypotheses are identified with a strong *a priori* prediction, the threshold of significance was set at $p < 0.05$, with small volume correction for multiple comparisons in the ROIs. To analyse these regression models, we will be using the MarsBaR toolbox (http://marsbar.sourceforge.net/) to anatomically define ROIs for the three prespecified brain regions (i.e. amygdala, vmPFC and left anterior insula) according to our main hypotheses. The GMVs will be

**Table 1.** Distribution of RWA and SDO. RWA, right-wing authoritarianism; SDO, social dominance orientation.

|  | total | | |
| --- | --- | --- | --- |
|  | mean | s.d. | range |
| RWA | −0.96 | 1.21 | (−4)–4 |
| SDO | 2.76 | 1.01 | 1–7 |

**Table 2.** Correlations between continuous self-report variables. RWA, right-wing authoritarianism; SDO, social dominance orientation.

|  | RWA | SDO | age |
| --- | --- | --- | --- |
| RWA | – | 0.453* | 0.478* |
| SDO |  | – | 0.189 |
| age |  |  | – |

*$p < 0.001$.

extracted from their ROIs using anatomically defined spheres with a radius of 20 mm centred at (MNI: $x = −36$, $y = −9$, $z = −17$) for the left amygdala and at (MNI: $x = 27$, $y = 12$, $z = −21$) for the right amygdala [60]. We then average the mean volumes from the left and right amygdala. The ROI of the vmPFC will be anatomically defined as a sphere with a radius of 20 mm centred at (MNI: $x = 0$, $y = 40$, $z = −18$; [91]). Finally, the ROI of the left anterior insula will be anatomically defined as a sphere with a radius of 20 mm centred at (MNI: $x = −45$, $y = 26$, $z = −6$; [67,68]). Similarly, age, gender and TIV will be included as covariates. After the ROI analyses, we will conduct Pearson's correlation tests for each ROI mean GMV with RWA and SDO separately. With regard to amygdala GMV, we average the volumes from the left and right amygdala to generate a single measure of bilateral amygdala volume.

# 3. Results

## 3.1. Self-report data

We first conducted a preliminary analysis on the RWA and SDO scales to ensure that they fulfil the assumptions necessary for parametric testing. None of the self-report items exceed the recommended cut-off values of ± 3, and therefore all items were included in the subsequent analyses. Both RWA and SDO scales demonstrated excellent internal reliability, $\alpha = 0.91$ and $\alpha = 0.92$, respectively. Table 1 shows the distribution of RWA and SDO by gender in the study.

### 3.1.1. In the social and political context of Singapore, are RWA and SDO scores correlated?

Table 2 shows the correlations between all self-report variables used in the study, excluding gender (since this is a categorical variable). Of note, the significant albeit moderate correlation between SDO and RWA ($r = 0.453$, $p < 0.001$) shows support for H1 (figure 1). We also ran an additional independent-sample $t$-test to determine existing gender differences between average RWA and SDO scores. Both RWA ($t = −1.718$, $p = 0.090$) and SDO ($t = −1.485$, $p = 0.142$) show no significant gender differences. Although age only correlated significantly with RWA ($r = 0.478$, $p < 0.001$) but not SDO ($r = 0.189$, $p = 0.089$), and gender did not show significant effect on RWA and SDO (both $p > 0.05$), these two variables were still included in the VBM analyses as covariates.

## 3.2. Whole brain analysis

As the main analysis, we conducted a whole-brain analysis with RWA and SDO as variables of interest. We included age, gender and TIV into the analysis model to regress out any effects of these covariates. First, we looked at what regions were positively associated with both RWA and SDO.

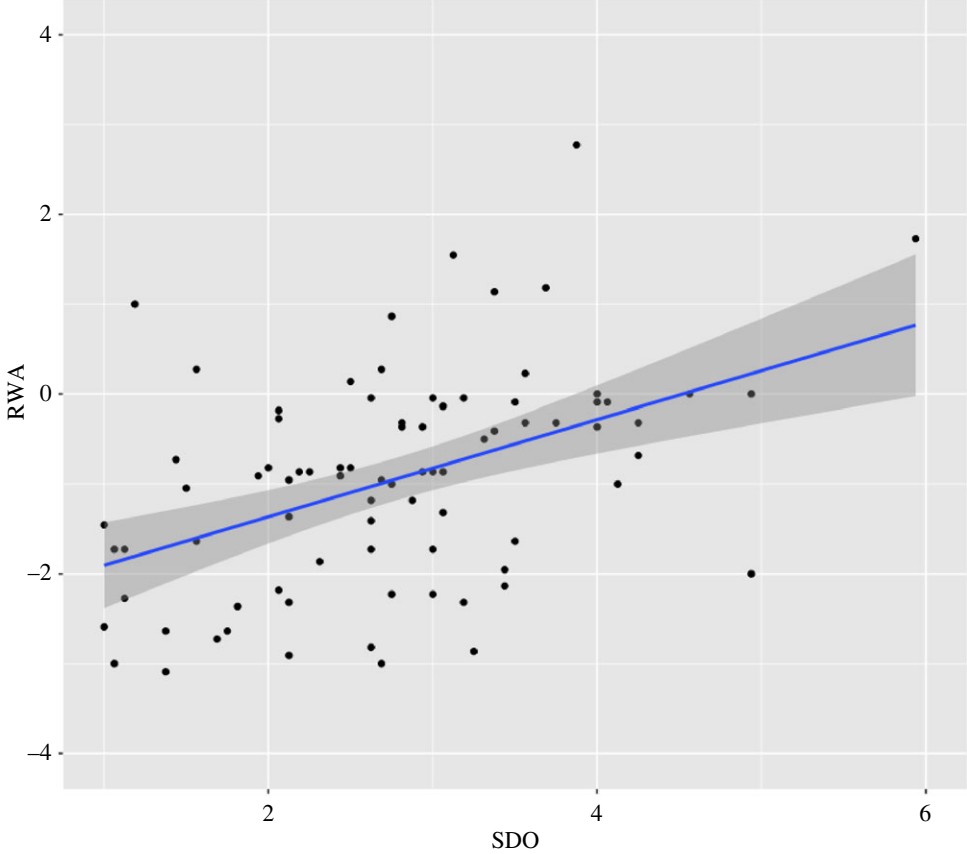

**Figure 1.** Scatter plot diagram of scores on social dominance orientation (SDO) scale by right-wing authoritarianism scale (RWA) with the best fit line (r = 0.453, $p < 0.001$).

### 3.2.1. Is/are there any overlapping region/s that are related to RWA and SDO?

Electronic supplementary material, figure S1 shows the glass brain images using uncorrected voxel-level analysis of regions positively associated with both RWA and SDO, with a minimum cluster of 20 voxels (see [60]). However, no clusters survived FWE correction ($p > 0.05$) (H2).

### 3.2.2. Is/are there any region/s that are related to RWA but not SDO? Is/are there any region/s that are related to SDO but not RWA?

Next, we looked at what regions were negatively associated with only RWA and, subsequently, those with only SDO. Electronic supplementary material, figure S2 shows the glass brain images using uncorrected voxel-level analysis of regions negatively associated with either RWA (H3) or SDO (H4), with a minimum cluster of 20 voxels. No clusters negatively associated with RWA (H3) or SDO (H4) survived FWE correction ($p > 0.05$).

## 3.3. Region of interest analysis

In addition to the whole-brain analysis, we also conducted *a priori* ROI analyses with RWA and SDO, as well as the same covariates to regress out their effects. This is to identify subtle neuroanatomical correlates as per our predictions that may have been missed in the whole-brain analyses. The stringent family-wise error corrections used in the latter are unlikely to yield significant clusters in a relatively small number of participants. Similarly, we used the criteria of 20 voxels minimum as a threshold to identify significant clusters.

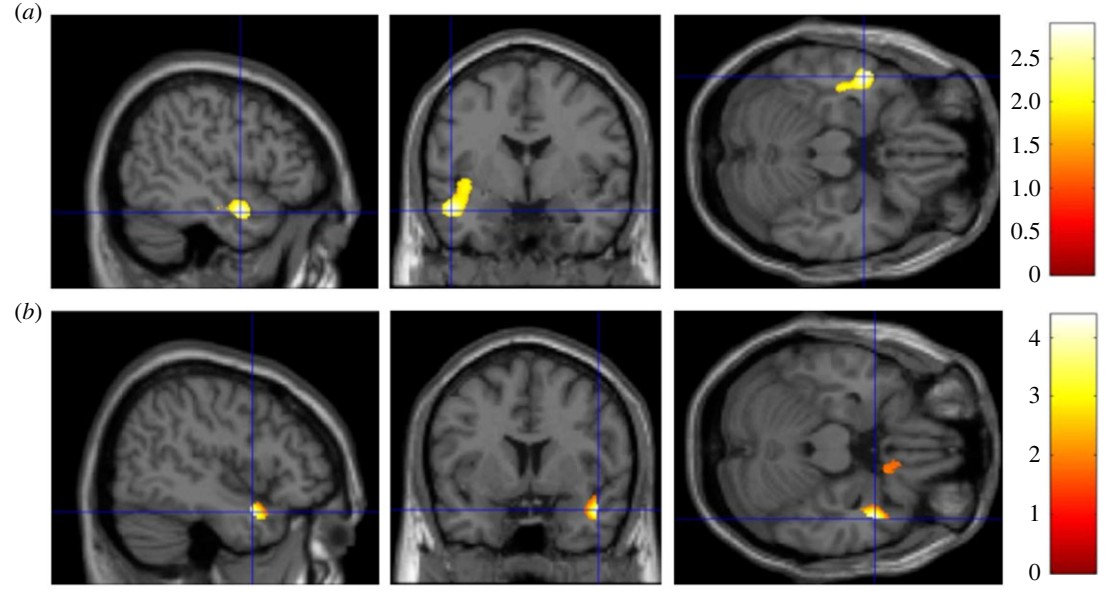

**Figure 2.** (*a*) Brain heatmap of a cluster within 20 mm radius of the left amygdala ($x = -49$, $y = -2$, $z = -21$) showing significant positive association with both RWA and SDO ($p < 0.05$). (*b*) Brain heatmap of two clusters within 20 mm radius of the right amygdala showing significant positive association with both RWA and SDO ($p < 0.05$).

### 3.3.1. Does grey matter volume in the amygdala relate to RWA and SDO?

In line with H2, RWA and SDO were both positively associated with GMV in a cluster within a 20 mm radius of the left amygdala ($t = 2.89$, $p = 0.003$, peak-level MNI coordinates: $x = -49$, $y = -2$, $z = -21$, figure 2*a*), and two clusters of the right amygdala ($p < 0.05$, figure 2*b*). Next, we looked at the association of RWA and SDO with vmPFC separately.

### 3.3.2. Does grey matter volume in the vmPFC relate to RWA but not SDO?

As predicted by H3, RWA was negatively associated with GMV in a cluster within 20 mm radius of the vmPFC ($t = 2.28$, $p = 0.013$, peak-level MNI coordinates: $x = -12$, $y = 31$, $z = -24$, figure 3*a*). However, SDO was also found to negatively associate with GMV in a cluster within 20 mm radius of the vmPFC ($t = 2.06$, $p = 0.021$, peak-level MNI coordinates: $x = -2$, $y = 56$, $z = -29$, figure 3*b*) which contradicts the prediction of H3.

### 3.3.3. Does grey matter volume in the left anterior insula relate to SDO but not RWA?

Finally, we looked at the association of RWA and SDO with the left anterior insula separately. As predicted by H4, SDO was negatively associated with GMV in two clusters within 20 mm radius of the left anterior insula ($p < 0.05$, figure 4). No significant clusters were found to be negatively associated with RWA ($p > 0.05$). Table 3 depicts all significant clusters ($k > 20$) within spherical ROIs associated with RWA and SDO.

## 3.4. Exploratory results

Due to the inherent size differences in the identified ROIs, spherical ROIs may not be appropriate to make direct comparison between these regions (i.e. amygdala, left anterior insula and vmPFC). We also should note that localization using spherical ROIs may identify significantly associated voxel clusters outside of the brain regions we sought to investigate. Therefore, in addition to our pre-registered findings using spherical ROIs, we conducted probabilistic atlas-based ROI analysis that classify structures into ROIs without assuming equivalency in size and circumscribe our analysis within established regional boundaries of the brain. We defined these ROIs using the Human Atlas of the WFU PickAtlas Tool (http://www.fmri.wfubmc.edu/cms/software#PickAtlas). With regard to the vmPFC, as the atlas did not have prespecified option for this region, we created a custom mask combining the following Brodmann areas used in Mackey & Petrides' [92] analysis of the architecture of the human vmPFC: 10, 11, 14, 24, 25, 32.

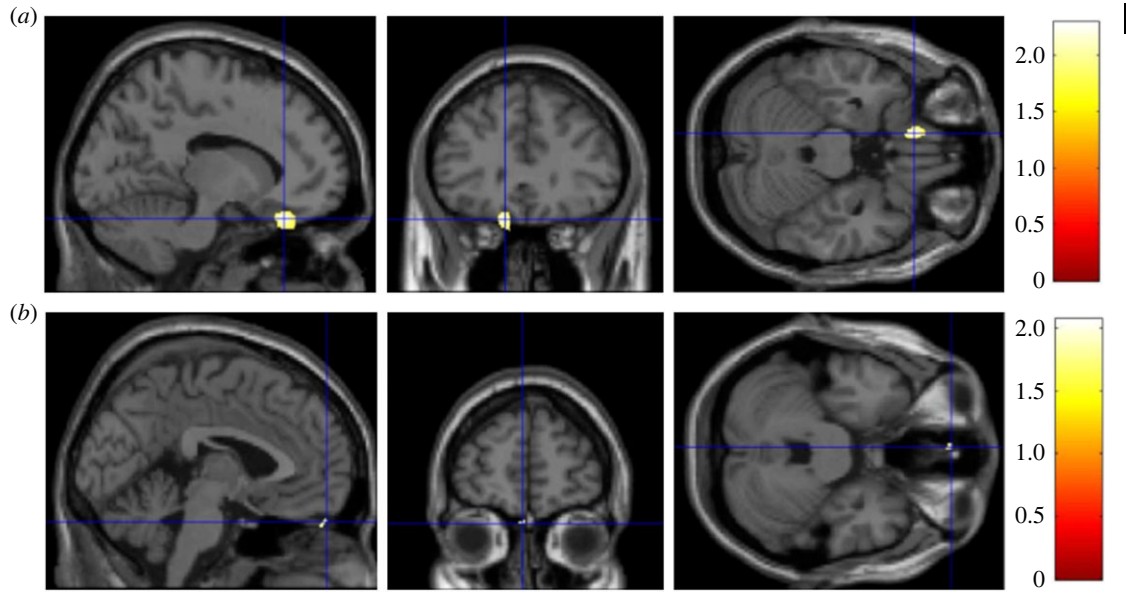

**Figure 3.** (*a*) Brain heatmap of a cluster within 20 mm radius of the vmPFC (*x* = −12, *y* = 31, *z* = −24) showing significant negative association with RWA (*p* < 0.05). (*b*) Brain heatmap of a cluster within 20 mm radius of the vmPFC (*x* = −2, *y* = 56, *z* = −29) showing significant negative association with SDO (*p* < 0.05).

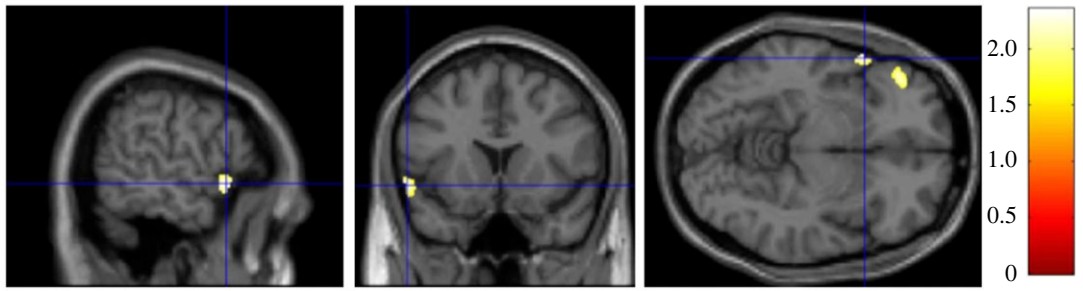

**Figure 4.** Brain heatmap of two clusters within 20 mm radius of the left anterior insula showing significant negative association with SDO (*p* < 0.05).

**Table 3.** Spherical ROI regions with GMV clusters associated with RWA and SDO. RWA, right-wing authoritarianism; SDO, social dominance orientation; vmPFC, ventromedial prefrontal cortex.

| contrasts (direction of predicted relationship) | region of interest | side | MNI coordinates | | | cluster size | *t*-value |
|---|---|---|---|---|---|---|---|
| | | | *x* | *y* | *Z* | | |
| RWA & SDO (positive) | amygdala | L | −49 | −2 | −21 | 3978 | 2.89 |
| | | R | 47 | 6 | −23 | 1258 | 4.37 |
| | | R | 15 | 13 | −27 | 626 | 2.16 |
| RWA (negative) | vmPFC | L | −12 | 31 | −24 | 671 | 2.28 |
| SDO (negative) | vmPFC | L | −2 | 56 | −29 | 53 | 2.06 |
| | anterior insula | L | −43 | 40 | −10 | 1036 | 2.25 |
| | | | −57 | 15 | −7 | 513 | 2.35 |

*p* < 0.05, uncorrected.

No significant clusters in the amygdala were found to be positively associated with RWA and SDO (*p* > 0.05), nor were clusters in the left anterior insula found to be negatively associated with RWA (*p* > 0.05) and SDO (*p* > 0.05), separately. One cluster in the vmPFC was found to be significantly negatively associated

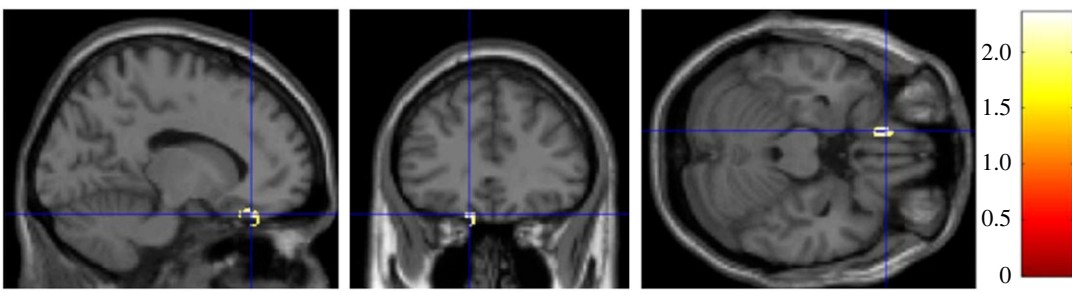

**Figure 5.** Brain heatmap of two clusters in vmPFC showing significant negative association with RWA ($p < 0.05$).

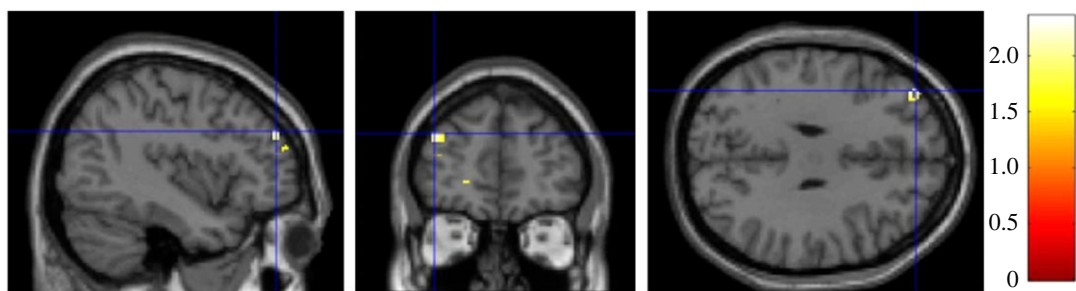

**Figure 6.** Brain heatmap of two clusters in vmPFC showing significant negative association with SDO ($p < 0.05$).

**Table 4.** Atlas-based ROI regions with GMV clusters associated with RWA and SDO. RWA, right-wing authoritarianism; SDO, social dominance orientation; vmPFC, ventromedial prefrontal cortex.

| contrasts (direction of predicted relationship) | region of interest | side | MNI coordinates | | | cluster size | t-value |
|---|---|---|---|---|---|---|---|
| | | | x | y | z | | |
| RWA (negative) | vmPFC | L | −13 | 33 | −23 | 367 | 2.26 |
| SDO (negative) | vmPFC | L | −40 | 47 | 30 | 520 | 2.52 |
| | | L | −24 | 51 | −1 | 87 | 2.15 |
| | | L | −4 | 68 | 12 | 91 | 2.14 |
| | | L | −44 | 40 | −12 | 51 | 2.11 |
| | | L | −20 | 48 | −1 | 30 | 2.07 |
| | | L | 3 | 68 | 11 | 26 | 2.03 |
| | | L | 12 | 70 | 14 | 40 | 2.00 |
| | | L | 31 | 68 | 7 | 22 | 2.00 |

$p < 0.05$, uncorrected.

with RWA ($t = 2.26$, $p = 0.013$, peak-level MNI coordinates: $x = −13$, $y = 33$, $z = −23$; figure 5) and multiple clusters significantly negatively associated with SDO ($p < 0.05$; figure 6). Table 4 depicts all significant clusters ($k > 20$) within atlas-based ROIs associated with RWA and SDO.

Additionally, the pre-registered findings were based on an arbitrary threshold value of 20 voxels to constitute a significant cluster in the brain associated with RWA or SDO. Further analysis using the threshold-free cluster enhancement (TFCE) toolbox [93] in SPM eliminated the need for using an arbitrary threshold value. No clusters survived ROI analysis using TFCE.

## 4. Discussion

The current study sought to identify significant neuroanatomical correlates of two system-justifying ideologies, namely, right-wing authoritarianism and social dominance orientation. Pearson's correlation test and VBM analyses were conducted to address four central hypotheses: (i) RWA and SDO will have a significant positive correlation, (ii) RWA and SDO will have a significant positive correlation with

amygdala volume, (iii) RWA will have a significant negative correlation with vmPFC, and (iv) SDO will have a significant negative correlation with left anterior insula. To our knowledge, this is the first study that has examined the structural neural substrates of RWA and SDO concurrently. The results demonstrate strong support for the first hypothesis predicting a significant positive correlation between RWA and SDO in a Singaporean sample. Whole brain analyses, after correction for multiple comparisons, yielded no clusters associated with either or both RWA and SDO. ROI analyses of the bilateral amygdala and vmPFC-identified clusters within 20 mm radius of these regions significantly associated with both RWA and SDO demonstrating support for hypothesis (ii) but did not support (iii). Finally, two clusters within 20 mm radius of the left anterior insula were significantly associated with SDO but no clusters were found to be associated with RWA, demonstrating support for hypothesis (iv).

We found that RWA and SDO within a Singaporean sample demonstrated a significant positive correlation. This runs contrary to the general finding in Roccato & Ricolfi's [24] cross-cultural meta-analysis where they concluded that individuals from industrialized nations characterized by low ideological contrasts such as Canada [1] would have, on average, RWA and SDO scores independent of each other. Singapore falls under this category, given that political attitudes and behaviour of its citizens do not neatly map onto a left–right dimension (e.g. liberal versus conservative). However, Singapore's large emphasis on multi-culturalism as reflected by its policies and institutional prescriptions (see [50]) is grossly understated in its influence on system-justifying ideologies (e.g. [40,94]). In Singapore, research has shown that one's support of local multi-culturalism policies is associated with RWA insofar as its ability to predict positive sentiments toward minority groups [25]. It would follow that the impact of multi-culturalism policies on SDO scores—which has been demonstrated in other country populations [53]—in Singapore would also lead to the direction of more positive outgroup attitudes. That is, RWA and SDO are likely to be shaped by multi-culturalism policies in a similar way. Additionally, Liu *et al.* [95] showed across five related studies that cultural-historical factors (i.e. charter status) have bidirectional influences on biculturalism, SDO and RWA in New Zealand and Taiwan. Taken altogether, these studies suggest that political factors moderate (and are moderated by) the relationship between RWA and SDO that are unaccounted for in the DPM model [46]. Factors associated with the political context (i.e. institutionalized multi-culturalism) we drew our sample from may explain the rather anomalous finding that the RWA and SDO scores of Singaporeans positively correlate with each other—running contrary to the predictions of the DPM model. Future research may benefit from incorporating direct measures of such political factors in the mix to empirically evaluate their underlying influence on system-justifying ideologies and substantiate these speculations.

The finding that GMV in bilateral amygdala is positively associated with both RWA and SDO is consistent with past research associating amygdala with system justification [60] and other interrelated variables such as outgroup discrimination [96] and perceived social ranking [97]. This finding lends support to the proposal that RWA and SDO are system-justifying ideologies. It has to do with scores in RWA and SDO scales sharing a neural correlate with scores in the general system justification (GSJ) scale [60]. Nam *et al.* argue that the GSJ is associated with amygdala volume, a brain region involved in the maintenance of social hierarchies [98], as the scale measures individual's propensity to defend an existing social system. It would follow that these two ideologies also espouse ways to legitimize inequalities in the status quo [21]. The importance of stable social hierarchies in humans can be seen by increased activity in the amygdala in situations perceived as unstable hierarchies [97]. The dlPFC was active in both stable and unstable hierarchy conditions probably because it simply was involved in assessing social status regardless of hierarchy stability [99]. By contrast, only in the unstable hierarchy condition was the amygdala activated owing to its role in processing threatening stimuli such as challenges to one's social rank [100]. Therefore, our results using spherical ROIs lends some support to the prediction that system-justifying ideologies, RWA and SDO, are related to individual differences in GMV within 20 mm radius of the amygdala, a brain region responsible for not only the maintenance of social hierarchies but also responses to stimuli threatening the stability of these hierarchies. Notwithstanding, our exploratory analysis using atlas-based ROIs found that the GMV in bilateral amygdala were not correlated with RWA or SDO. It is likely that the localization method (i.e. spherical ROIs) may have inadvertently identified voxels beyond the amygdala, and hence, we exercise caution in making any strong interpretation based on the pre-registered analysis.

We also found that GMV within 20 mm radius of the vmPFC was negatively associated with both RWA and SDO. The link between vmPFC GMV and RWA scores is consistent with past research showing ablations of the vmPFC is directly associated with increase in RWA scores [61]. When compared with control participants, patients with vmPFC lesions have a larger tendency to be misled by false information (e.g. advertisements; [65]). This connection between vmPFC and RWA can be explained by

FTT theory, whereby beliefs are developed in two stages: (i) information is represented in a mental space, and (ii) information is assessed for falsehoods (see [62]). The vmPFC is implicated in this second stage whereby damage to this region puts a brake on one's ability to correctly identify wrong information. This has broad implications on the biological mechanism that underlies authoritarians' view towards societal issues such as abortion and drug use [63], and its relation with fundamentalist beliefs [64]. Interestingly, SDO was also shown to be negatively associated with vmPFC GMV. The vmPFC is associated with many functions ranging from emotional regulation [101] to economic valuation [102], which unsurprisingly correlate with RWA [103,104]. This range of domains can be subsumed under the common function of affective appraisal of outcomes based on conceptual information [105]. Individuals with high RWA, by its definition, glean information from authority figures and in turn influence emotional responses to, say, outgroup members [1], whereas the emotional response of individuals with high SDO towards outgroup members is driven by information on relative superiority of another group [47]. In the latter case, vmPFC may be associated with SDO because the emotional response of high SDO scorers is also motivated by information, albeit for a different motivation (i.e. dominance over others). Though this does not necessarily contradict the tenets of the DPM in that these two ideologies with different underlying motivations ought to be associated with different brain regions (in addition to overlapping brain regions, e.g. amygdala), the current study cannot draw any definitive conclusions regarding non-overlapping brain regions linked with only RWA but not SDO. Future work may look into brain regions associated with variables theoretically preceding RWA and SDO on the causal DPM model [46]. For instance, research has shown that brain regions independently associate with different personality traits, like agreeableness and openness to experience [106], which roughly correspond to social conformity and tough-mindedness traits that motivate RWA and SDO, respectively [22]. It is worth noting that both registered and exploratory analysis using spherical and atlas-based ROIs, respectively, found clusters in the vmPFC associated with RWA and SDO. One major contributing factor for this is that the vmPFC constitute a relatively large portion of the cortex [92], as compared with the amygdala and left anterior insula.

Finally, we also found that GMV within 20 mm radius of the left anterior insula region was negatively associated with SDO but not with RWA. This is evidence to suggest that activity in this region during empathy-related tasks is moderated by SDO [68,107]. The insula is a major player in the empathy network [108]. Even after controlling for dispositional empathy, SDO moderated the activity of the left anterior insula when observing others experiencing pain stimuli, suggesting that SDO has a distinct relationship with empathic concern beyond one's dispositional trait [68]. Past research has also demonstrated that SDO moderates activity in the insula beyond empathy-related tasks. Cazzato *et al*. [67] found SDO scores moderated activity in the social orienting circuit, which includes the left anterior insula when following gaze direction in images of political figures. The observation that SDO moderates left anterior insula activity in empathy-related and low-level attention-related tasks points to a more direct association between SDO and the left anterior insula. One potential explanation would be the insula's role in aversion towards social inequity [109]. The negative association between SDO and the left anterior insula then becomes clearer as the SDO ideology promotes stability of a hierarchical social structure [21]. That is, individuals with higher SDO scores are likely to have smaller insula volumes and presumably, lower aversion towards social inequity. Conversely, RWA scores were not associated with left anterior insula GMV as was predicted. High RWA scorers may not necessarily be concerned with social inequity that results from dominance of one group over another as much as they are with threats to the stability of the overall social structure, such as socially deviant behaviour [47]. This is consistent with the underlying motivations of RWA and SDO predicted by the DPM model (i.e. perceived threat versus competition; [22]). Notwithstanding, these are speculations that should be qualified by more direct tests of the role of SDO (and RWA) on attitudinal measures (see experimental design in [110]). As with the amygdala, atlas-based ROI localization found no correlation between GMV in the left anterior insula with RWA or SDO. Therefore, the significant clusters identified by the spherical ROI localization method associated with SDO may not necessarily constitute volume in the left anterior insula.

# 5. Limitations and future directions

The current study has shed light on the neural substrates underlying SDO and RWA, but this is not without limitations. First, the relatively small sample size may be too small for a whole-brain analysis with stringent statistical corrections for multiple comparison to detect an effect. Moreover, a larger sample will be

required to improve the power and, consequently, the reproducibility of such findings [111]. Second, cultural factors associated with the sample ethnicity should be accounted for in future work investigating both behavioural and neural correlates of system-justifying ideologies. With regard to our sample of ethnically Chinese individuals, Liu *et al.* [112] showed a strong association of guanxi—a set of beliefs commonly encouraging social networks and relationships to be used to facilitate social transactions—with both RWA and SDO. Third, this study only sampled an ethnically Chinese sample, therefore obviating any generalization of these neuroanatomical correlates to the general Singapore population. Though the multi-culturalism policies are enforced relatively uniformly across Singapore, the psychological and underlying neural effects of such policies may be different for ethnic minorities in Singapore [51]. Future research may also consider directly comparing national populations on different points of the ideological contrast spectrum to determine if the neuroanatomical correlates of RWA and SDO vary as a function of a nation's political context. Fourth, as one of the key frameworks of this study implicitly rests on the system justification theory, future work should also include the GSJ scale [90] as a potential baseline for identifying the neural substrates of specific system-justifying ideologies, including RWA, SDO, meritocracy and so on. Finally, an important caveat must be made regarding findings based on our pre-registered methodology: using relatively large spherical ROIs (i.e. radius of 20 mm) may have mischaracterized significant clusters into our ROIs, particularly small-volume structures such as the amygdala. Our exploratory findings using atlas-based ROIs precludes any categorical conclusion based on the present evidence.

## 6. Conclusion

In summary, the results show that in a Chinese Singaporean sample, RWA and SDO are strongly correlated given the low ideological contrast of Singapore. This is probably due to the country's proactive stance on institutionalized multi-culturalism which has an undeniable influence on a range of environments from housing to schools to the workplace. The two-dimensional evolutionary framework and DPM model offer promising theoretical bases to inform the neuroscientific research on system-justifying ideologies (e.g. RWA, SDO). Though the whole brain found no significant regions associated with these ideologies, ROI analyses in this study identified overlapping (i.e. amygdala and vmPFC) and non-overlapping (i.e. left anterior insula) regions associated with RWA and SDO for future replication with specific call for atlas-based ROI analysis. Our exploratory analysis, however, suggests that our spherical ROI localization approach may have identified clusters that were not necessarily within the amygdala and left anterior insula. Nevertheless, the findings of this study demonstrate the viability of hypothesis-driven political neuroscience to advance our knowledge of how humans draw on ideologies to make sense of existing social structures.

**Ethics.** Each participant was required to provide written informed consent before participating in the study in accordance to the Declaration of Helsinki. This study was approved by the NTU IRB (Protocol 2017-01-029).

**Data accessibility.** This project was pre-registered on the open-science framework (OSF) at http://osf.io/9aebg. The repository also contains a csv file enclosing data to reproduce the ROI analyses, including nuisance variables (i.e. gender, age, TIV) and variables of interest (i.e. RWA, SDO). All raw neuroimaging data can be found in the DR-NTU repository for both the first (doi:10.21979/N9/H0CPDV) and second (doi:10.21979/N9/BFZ26X) recruitment phases.

The data are provided in electronic supplementary material [113].

**Authors' contributions.** J.P.M.B.: conceptualization, formal analysis, investigation, methodology, writing—original draft; S.T.: conceptualization, supervision, writing—review and editing; B.L.R.: supervision, writing—review and editing; P.R.: supervision; M.H.B.: supervision; G.E.: conceptualization, project administration, supervision, writing—review and editing.

All authors gave final approval for publication and agreed to be held accountable for the work performed therein.

**Conflict of interest declaration.** The authors declare that they have no known competing financial interests or personal relationships that could have appeared to influence the work reported in this paper.

**Funding.** This research was supported by grants from the Ministry of Education, Singapore, under its Academic Research Fund Tier 1 (grant nos. RT10/19 and RG55/18). The funders had no role in study design, data collection and analysis, decision to publish or preparation of the manuscript.

**Disclosure.** Dr Tolomeo has received unrestricted educational grants from Indivior, Lundbeck and Merck Serono. This study was pre-registered at Peer Community In Registered Reports (https://rr.peercommunityin.org/) and the Pre-registered Stage 1 protocol was posted on 19 August 2021 (Recommendation: posted 16 March 2022, validated 16 March 2022), while the Pre-registered Stage 2 was posted on 27 July 2022 (Recommendation: posted 9 February 2023, validated 10 February 2023). All the information can be retrieved at: https://osf.io/btkwq and https://rr.peercommunityin.org/articles/rec?id=263.

# Appendix A

Items on right-wing authoritarianism scale

Participants were instructed to rate their agreement or disagreement with each statement from −4 (strongly disagree) through 0 (neutral) to +4 (strongly agree).

1. The established authorities generally turn out to be right about things, while the radicals and protestors are usually just 'loud mouths' showing off their ignorance.
2. Women should have to promise to obey their husbands when they get married.
3. Our country desperately needs a mighty leader who will do what has to be done to destroy the radical new ways and sinfulness that are ruining us.
4. Gays and lesbians are just as healthy and moral as anybody else.*
5. It is always better to trust the judgement of the proper authorities in government and religion than to listen to the noisy rabble-rousers in our society who are trying to create doubt in people's minds.
6. Atheists and others who have rebelled against the established religions are no doubt every bit as good and virtuous as those who attend church regularly.*
7. The only way our country can get through the crisis ahead is to get back to our traditional values, put some tough leaders in power, and silence the troublemakers spreading bad ideas.
8. There is absolutely nothing wrong with nudist camps.*
9. Our country needs free thinkers who have the courage to defy traditional ways, even if this upsets many people.*
10. Our country will be destroyed someday if we do not smash the perversions eating away at our moral fibre and traditional beliefs.
11. Everyone should have their own lifestyle, religious beliefs and sexual preferences, even if it makes them different from everyone else.*
12. The 'old-fashioned ways' and the 'old-fashioned values' still show the best way to live.
13. You have to admire those who challenged the law and the majority's view by protesting for women's abortion rights, for animal rights, or to abolish school prayer.*
14. What our country really needs is a strong, determined leader who will crush evil, and take us back to our true path.
15. Some of the best people in our country are those who are challenging our government, criticizing religion, and ignoring the 'normal way things are supposed to be done'.*
16. God's laws about abortion, pornography and marriage must be strictly followed before it is too late, and those who break them must be strongly punished.
17. There are many radical, immoral people in our country today, who are trying to ruin it for their own godless purposes, whom the authorities should put out of action.
18. A 'woman's place' should be wherever she wants to be. The days when women are submissive to their husbands and social conventions belong strictly in the past.*
19. Our country will be great if we honour the ways of our forefathers, do what the authorities tell us to do, and get rid of the 'rotten apples' who are ruining everything.
20. There is no 'ONE right way' to live life; everybody has to create their own way.*
21. Homosexuals and feminists should be praised for being brave enough to defy 'traditional family values'.*
22. This country would work a lot better if certain groups of troublemakers would just shut up and accept their group's traditional place in society.

*items to be reverse-coded before calculating total RWA score

# Appendix B

Items on the social dominance orientation scale

Participants were instructed to rate their agreement or disagreement with each statement from 1 (strongly disagree) to 7 (strongly agree).

1. Some groups of people are simply inferior to other groups.
2. In getting what you want, it is sometimes necessary to use force against other groups.
3. It's OK if some groups have more of a chance in life than others.
4. To get ahead in life, it is sometimes necessary to step on other groups.
5. If certain groups stayed in their place, we would have fewer problems.

6.  It's probably a good thing that certain groups are at the top and other groups are at the bottom.
7.  Inferior groups should stay in their place.
8.  Sometimes other groups must be kept in their place.
9.  It would be good if groups could be equal.*
10. Group equality should be our ideal.*
11. All groups should be given an equal chance in life.*
12. We should do what we can to equalize conditions for different groups.*
13. Increased social equality is beneficial to society.*
14. We would have fewer problems if we treated people more equally.*
15. We should strive to make incomes as equal as possible.*
16. No group should dominate in society.*

*items to be reverse-coded before calculating total SDO score

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
