## [Peer Review File · Royal Society Open Science]

Review History

Decision letter (RSOS-230196.R0)

Dear Dr Esposito

Per our earlier correspondence in relation to this Stage 1 Registered Report submitted from PCI Registered Reports, the journal team will now submit the Stage 2 Registered Report on your behalf.

Nevertheless, find below, for your records, the Stage 1 in-principle acceptance.

With my best
Andrew Dunn

===

On behalf of the Editor, I am pleased to inform you that your Manuscript RSOS-230196 entitled "Pre-registered Voxel-based Morphometry Study on Right-Wing Authoritarianism and Social Dominance Orientation" has been accepted in principle for publication in Royal Society Open Science. The reviewers' and editors' comments are included at the end of this email.

You may now progress to Stage 2 and complete the study as approved. Before commencing data collection we ask that you:

- 1) Update the journal office as to the anticipated completion date of your study.
- 2) Register your approved protocol on the Open Science Framework (<https://osf.io/>) or other recognised repository, either publicly or privately under embargo until submission of the Stage 2 manuscript. Please note that a time-stamped, independent registration of the protocol is mandatory under journal policy, and manuscripts that do not conform to this requirement cannot be considered at Stage 2. The protocol should be registered unchanged from its current approved state, with the time-stamp preceding implementation of the approved study design.

Following completion of your study, we invite you to resubmit your paper for peer review as a Stage 2 Registered Report. Please note that your manuscript can still be rejected for publication at Stage 2 if the Editors consider any of the following conditions to be met:

- The results were unable to test the authors' proposed hypotheses by failing to meet the approved outcome-neutral criteria.
- The authors altered the Introduction, rationale, or hypotheses, as approved in the Stage 1 submission.
- The authors failed to adhere closely to the registered experimental procedures. Please note that any deviations from the approved experimental procedures must be communicated to the editor immediately for approval, and prior to the completion of data collection. Failure to do so can result in revocation of in-principle acceptance and rejection at Stage 2 (see complete guidelines for further information).
- Any post-hoc (unregistered) analyses were either unjustified, insufficiently caveated, or overly dominant in shaping the authors' conclusions.
- The authors' conclusions were not justified given the data obtained.

We encourage you to read the complete guidelines for authors concerning Stage 2 submissions at <https://royalsocietypublishing.org/rsos/registered-reports#ReviewerGuideRegRep>. Please especially note the requirements for data sharing, reporting the URL of the independently registered protocol, and that withdrawing your manuscript will result in publication of a Withdrawn Registration.

Your feedback matters - please spend 5 minutes leaving anonymous feedback about your experience of Registered Reports at this journal, as an author or reviewer: https://registeredreports.cardiff.ac.uk/feedback/feedback/decision_letter.php

This feedback is collected by the Registered Reports Community Feedback, website which is an independent service and research project, being undertaken by Cardiff University.

Once again, thank you for submitting your manuscript to Royal Society Open Science and we look forward to receiving your Stage 2 submission. If you have any questions at all, please do not hesitate to get in touch. We look forward to hearing from you shortly with the anticipated submission date for your stage two manuscript.

on behalf of Professor Chris Chambers (Associate Editor) and Chris Chambers (Registered Reports Editor, Royal Society Open Science)
openscience@royalsociety.org

Associate Editor Comments to Author (Professor Chris Chambers):

Associate Editor: 1

Comments to the Author:

(There are no comments.)

Associate Editor: 2

Comments to the Author:

(There are no comments.)

Reviewers' comments to Author:

Author's Response to Decision Letter for (RSOS-230196.R0)

See Appendix A.

Decision letter (RSOS-230196.R1)

Dear Dr Esposito:

I am pleased to inform you that your manuscript entitled "Pre-registered Voxel-based Morphometry Study on Right-Wing Authoritarianism and Social Dominance Orientation" is now accepted for publication in Royal Society Open Science.

Please ensure that you send to the editorial office an editable version of your accepted manuscript, and individual files for each figure and table included in your manuscript. You can send these in a zip folder if more convenient. Failure to provide these files may delay the processing of your proof.

on behalf of Professor Chris Chambers (Subject Editor).

Follow Royal Society Publishing on Twitter: @RSocPublishing
Follow Royal Society Publishing on Facebook:
<https://www.facebook.com/RoyalSocietyPublishing/>
Read Royal Society Publishing's blog:
<https://royalsociety.org/blog/blogsearchpage/?category=Publishing>

Appendix A

This Registered Report was submitted to *Royal Society Open Science* following peer review and recommendation for Stage 2 acceptance at the Peer Community In (PCI) Registered Reports platform. Full details of the peer review and recommendation of the paper at PCI Registered Reports may be found at the links described below.

After submission to the journal, the paper received no additional external peer review, but was accepted on the basis of the Editor's recommendation according to our PCI Registered Reports policy <https://royalsocietypublishing.org/rsos/registered-reports#PCIRR>.

Link to the Stage 1 recommendation and review history at PCI Registered Reports
<https://rr.peercommunityin.org/articles/rec?id=97>

Link to the Stage 2 recommendation and review history at PCI Registered Reports
<https://rr.peercommunityin.org/articles/rec?id=263>